# Role of Circulating Tumor DNA in Gastrointestinal Cancers: Current Knowledge and Perspectives

**DOI:** 10.3390/cancers13194743

**Published:** 2021-09-22

**Authors:** Emilie Moati, Valerie Taly, Simon Garinet, Audrey Didelot, Julien Taieb, Pierre Laurent-Puig, Aziz Zaanan

**Affiliations:** 1Department of Gastroenterology and Digestive Oncology, Institut du Cancer Paris Carpem, Assistance Publique des Hôpitaux de Paris, European Georges Pompidou Hospital, 75015 Paris, France; emilie.moati@aphp.fr (E.M.); julien.taieb@aphp.fr (J.T.); 2Centre de Recherche des Cordeliers, INSERM UMRS1138, Centre National de la Recherche Scientifique, Sorbonne Université, USPC, Université de Paris, Equipe Labellisée Ligue Nationale Contre le Cancer, CNRS SNC 5096, 75006 Paris, France; valerie.taly@parisdescartes.fr (V.T.); simon.garinet@aphp.fr (S.G.); audrey.didelot@parisdescartes.fr (A.D.); pierre.laurent-puig@parisdescartes.fr (P.L.-P.); 3Department of Biochemistry, Institut du Cancer Paris Carpem, Assistance Publique des Hôpitaux de Paris, European Georges Pompidou Hospital, 75015 Paris, France

**Keywords:** circulating tumor DNA, gastrointestinal cancers, personalized medicine, biomarker

## Abstract

**Simple Summary:**

Management of gastrointestinal (GI) cancers t is a worldwide challenge and some new tools are needed to guide it. Circulating tumor DNA (ctDNA) is a fraction of tumor DNA that can be detected by a liquid biopsy through a simple blood sample. In this work, we tried to summarize in a clinical review to what extend the analysis of ctDNA can improve therapeutic support in digestive oncology and how this circulating biomarker appears as a very promising improvement in addition to classic diagnostic, prognostic and theranostic methods. Although the level of evaluation of this tool is still different between the different GI cancers locations, it is in dynamic evolution in all of them.

**Abstract:**

Gastrointestinal (GI) cancers are major health burdens worldwide and biomarkers are needed to improve the management of these diseases along their evolution. Circulating tumor DNA (ctDNA) is a promising non-invasive blood and other bodily-fluid-based biomarker in cancer management that can help clinicians in various cases for the detection, diagnosis, prognosis, monitoring and personalization of treatment in digestive oncology. In addition to the well-studied prognostic role of ctDNA, the main real-world applications appear to be the assessment of minimal residual disease to further guide adjuvant therapy and predict relapse, but also the monitoring of clonal evolution to tailor treatments in metastatic setting. Other challenges such as predicting response to treatment including immune checkpoint inhibitors could also be among the potential applications of ctDNA. Although the level of advancement of ctDNA development in the different tumor localizations is still inhomogeneous, it might be now reliable enough to be soon used in clinical routine for colorectal cancers and shows promising results in other GI cancers.

## 1. Introduction

Gastrointestinal (GI) cancers appear as major health burdens worldwide with high incidences and mortality rates. For these cancers, stage at diagnosis remains the most important prognostic factor for clinical outcome. However, the emergence of simple and reproducible biomarkers is needed for the management of these diseases along their evolution. Circulating cell-free DNA (cfDNA) can be detected in plasma, urine, and other bodily fluids for everyone, and is increased in inflammatory diseases, infections and cancers [1,2]. For patients with cancer, a fraction of this cfDNA, called circulating tumor DNA (ctDNA), contains tumor-specific molecular alterations [3,4]. Detection of ctDNA is challenging: First, for the majority of patients, quantities remain very low. Moreover, ctDNA is diluted within total cfDNA and its identification can be difficult. New approaches aretherefore in development to overcome this sensitivity challenge. Depending on cancers subtypes, specific molecular alterations can attest for the presence of ctDNA, which is a promising non-invasive biomarker in the era of personalized medicine. In this review, we tried to resume the molecular aspects of ctDNA and in what extent this biomarker can help clinicians in the detection, screening, diagnosis, prognosis, monitoring and personalization of treatment in patients with gastrointestinal cancers.

## 2. Molecular Aspects

The first challenge in detection of ctDNA was due to the low quantities of DNA extracted from plasma. While amounts of cfDNA are higher in plasma from patients with cancer, ranges of elevation remain around an average of 3–4 times the quantity of a healthy individual [5]. However, the most challenging technical aspect is the detection of the low ratio of ctDNA representing 1 to 10% of cfDNA for advanced stages and until <0.1% particularly in early stages [5] (Figure 1).

The 1% allelic ratio threshold is commonly considered as the limit of detection for conventional quantitative polymerase chain reaction (PCR) technologies [6]. Recent advances in sequencing technologies allowed to detect these rare mutations within a background of wild type sequences, and to screen multiple genomic regions in a single run. The highly sensitive techniques that are required for liquid biopsy analyses can be broadly classified as digital PCR-based approaches and massive multiplexing next-generation sequencing (NGS)-based approaches [7].

Digital PCR (dPCR) techniques can identify specific known tumor mutations with high accuracy. Indeed, partitioning DNA molecules through the generation of millions of droplets and the possible additivity of reactions reduce template competition and allows to reach a theoretical detection threshold of mutations until 1/100,000 [8]. Therefore, digital droplet PCR (ddPCR) might appear as the most sensitive, suitable and fast technology [7,9,10,11]. Several ddPCR platforms are now being commercialized but still contain a restricted number of probes in each run. Therefore, their use in clinical routine remains limited to the detection of specific pre-known mutations [10].

NGS provides a comprehensive profile of molecular alterations occurring along tumor evolution, without necessity of their prior knowledge. First common NGS panels permitted to reach a detection threshold around 1–2%, stressing the need to improve their sensitivity for relevant liquid biopsies applications. Indeed, sequencing abilities are limited by several issues: depth of sequencing, background error noise and the methodological pipeline used to detect mutations.

Then, bioinformatics methods were developed to improve pipelines initially designed for tissue analysis. For example, Base Position Error Rate (BPER) is an algorithm based on the fact that 1–2% of error rate reported by NGS manufacturers refers to a mean of all type of error rates, which are highly variable along the genome because of sequencing environment. In this technique, a workflow recalculates an error rate for each position to be able to identify mutations at 0.3% ratio for single nucleotide variants and at 0.1% for indel >2 [12]. This algorithm provides a cost-effective solution to identify mutations in easily sequenced regions but however does not reach dPCR sensitivity.

Therefore, NGS sequencing technologies were upgraded to better discriminate a real variant from background noise. They can be divided in 3 categories: (1) capture of ctDNA with specific probes, (2) PCR amplification of selected regions, with short amplicons size (<150 bp), (3) anchored multiplex PCR, which theoretically enriches for highly fragmented ctDNA over high molecular weight genomic DNA. These technologies are frequently coupled with unique molecular identifiers (UMI), an initial random barcoding with short (8–16 bp) DNA sequences tagging each individual DNA fragment before amplification [13]. However, the required sequencing depth and the cost are more elevated. Elazezy et al. developed and compared various methods although they are not easily manageable by clinical laboratories yet [14]. Numerous commercial kits for ctDNA libraries are therefore available, with a range of targeted genes from 20 to >500 and using the UMI technology. Their heterogeneity of design and performances complicates the choice for clinical laboratories. Therefore, their evaluation and comparison in real life are strongly mandatory [15,16].

Another currently explored filed of interest is exosomes’ concentration and sequencing as these vesicles are actively excreted by tumor cells. However, in a recent cohort of 33 CRC, Thakur et al. did not report any improvement in the sensitivity for ctDNA detection [17].

Beyond the detection of point mutations, other technologies are now available through the analysis of liquid biopsy.

Shen et al. confirmed robust performance in universal cancer detection and classification across an extensive collection of plasma samples from several tumor types based on cell-free methylation patterns [18]. Recent validation results of the Circulation cell free Genome atlas study based on methylation and machine learning showed a specificity of 99.5% and a sensitivity of 50% for cancer detection [19].

More recently, Mathios et al. evaluated the cell free fragmentomes instead of ctDNA and reported good performances to detect lung cancer even at early stages (91% of stages I/II, and 96% of stages III/IV), thus opening a new paradigm for liquid biopsy [20].

Finally, other approaches are currently emerging to overcome sequencing technologies biases, such as plasmonic nanoparticles used in Surface-Enhanced Raman Scattering (SERS), mass spectrometry based assays and electrochemical biosensor technologies [7,21,22,23].

## 3. Early Cancer Detection through Circulating Tumor DNA and Molecular Profile Determination

The GI cancer diagnosis is currently based on a histological assessment and therefore requires tissue sample collected by surgical resection, endoscopic ultrasound, or biopsy of primitive tumor or accessible metastasis. Several studies assessed the interest of ctDNA as screening tool for early tumor stage. However, further studies are still required to prove the clinical utility of ctDNA in early diagnosis as stipulated by American Society of Clinical Oncology (ASCO) and the College of American Pathologists (CAP) in a recent report [24]. Some findings demonstrated that asymptomatic cancers could be detected years before conventional diagnosis through non-invasive blood tests. In a recent longitudinal study, analysis of ctDNA methylation was performed on plasma samples from 605 asymptomatic individuals. Among them, 191 later developed stomach, esophageal, colorectal, lung or liver cancer within four years of blood draw. This method was able to detect cancer in 95% of asymptomatic individuals who were later diagnosed [25]. However, future longitudinal studies are required to confirm these results. The main risk of this early screening would be over-diagnosis through false-positive results or through the detection of circulating genomic variants from cells that have taken the first step toward transformation but were never meant to become clinically important [24].

Tissue biopsy is usually only performed at diagnosis and can sometimes be hard to obtain. For these reasons, several studies have also evaluated in different GI cancers whether plasma molecular alterations can be detected with ctDNA and are correlated with tissue biopsies.

### 3.1. Colorectal Cancer

Tumor tissue is routinely used to search for *KRAS* or *NRAS* gene mutations that occur in around 55% of metastatic CRC (mCRC) and predict a lack of response to the EGFR-targeted monoclonal antibodies, such as cetuximab and panitumumab [12,26]. In the same context, BRAF mutation is another alteration known as a poor prognostic factor that can be targeted by a doublet-therapy combining an anti-BRAF kinase inhibitor (encorafenib) and anti-EGFR monoclonal antibody (cetuximab) [27,28,29].

In the context of mCRC, the quantitative PCR (Intplex qPCR) on ctDNA was described by Thierry et al. as a valuable detection method with a high rate of specificity and sensitivity, especially for *BRAF* V600E and *KRAS* mutations, in a prospective study on 106 patients with mCRC [30]. The digital droplet PCR (ddPCR) has also been validated by other group for detection of *KRAS* mutations in mCRC [31]. More recently, in a large prospective multicenter study. Another method consists in using the NGS-BEPER-method (22 genes), and two specific methylated biomarkers (*WIF1* and *NPY*) as a second-step test for NGS-negative specimens. Bachet et al. used this technique to evaluate the concordance of *RAS* mutations between plasma and tissue among 406 chemotherapy-naive patients with mCRC with detectable ctDNA (*n* = 329/412). By comparing the results of RAS status in ctDNA and in matched tumor tissus, they founded an accuracy of 83% with NGS alone versus 93% with NGS plus methylated biomarkers [32]. Supplementary studies also suggested a good concordance rate between mutations observed in tumor biopsy and those identified on ctDNA [30,31,32,33,34,35,36].

### 3.2. Pancreatic Cancer

The *KRAS* gene mutations occur in more than 90% of pancreatic cancer (PC), and appears therefore as the best candidate to assess the presence of ctDNA in this tumor [37,38,39,40]. However, the ctDNA detection rate in metastatic PC varies widely from 40% to 80% and could therefore explain some discordance between tumor and plasma mutation assessment [41,42,43,44]. It could explain the results of the recent meta-analysis of Luchini et al. including 14 studies involving 369 patients, that reported a concordance rate of only 32% between ctDNA and tissue based on large NGS multi-gene mutation panels [45]. The overall pooled sensitivity and specificity of the mutational analysis on liquid biopsy compared to tumor tissue were 70% and 86% respectively. However, when focusing on studies analyzing *KRAS* mutations only, the sensitivity slightly decreased but the specificity increased and were 65% and 91%, respectively [45].

Indeed, apart from *KRAS* mutations for PC screening, adding NGS-based panel for other mutations such as *SMAD4, CDKN2A, ROS1, BRAF* and *TP53* could lead to higher levels of ctDNA detection [46,47,48]. More recently, methylation of promoter of *ADAMST1* and *BNC1* genes were also described as potential tool to assess the presence of ctDNA in PC [49].

However, the use of highly sensitive detection methods of ctDNA might lead to false diagnosis of PC. Indeed, *KRAS* mutations can be detected in plasma in some non-cancerous diseases such as chronic pancreatitis. In a pilot study from Rashid et al., 21.8% of patients with chronic pancreatitis were tested positive for *KRAS* mutations in plasma [50]. Among these 64 patients, none developed a PC, with a mean follow-up duration (by clinic and by positron emission tomography or endoscopic ultrasound) of 2.5 years [50].

Quantitative ctDNA assessment, or combining biomarkers and methylation detection may improve the specificity of ctDNA detection and therefore help to discriminate benign from malignant pancreatic diseases, even at early tumor stages [51,52,53,54].

### 3.3. Esophageal and Gastric Cancer

In gastric cancer (GC), despite a low-frequency of genomic alterations [55,56], routine tissue-based NGS showed that at least 37% of patients harbor somatic mutations (*TP53, KRAS*) or gene amplification, such as *HER2, MET, EGFR,* and *FGFR2* [57,58,59,60]. Some retrospective studies evaluated the feasibility of ctDNA detection by NGS among GC patients. In a recent study including 55 patients with GC tested by NGS, Kato et al. showed that 31 had concordant mutations between tumor tissue and ctDNA with levels ranged from 61.3% (for *TP53* mutation) to 87.1% (for *KRAS* mutation) [61]. In their meta-analysis, Gao et al., reported that ctDNA detection might be a specific, but still a low sensitive test in GC patients [62]. More recently, the analysis of a large cohort of 1630 patients with GC revealed that ctDNA-NGS genomic landscape was similar but not identical to tissue-NGS [63]. This could reflect the molecular heterogeneity, with some targetable molecular alterations identified at higher frequency via ctDNA-NGS compared with previous matched primary tissue-NGS samples [63].

Despite increasing use of genomic alterations to detect ctDNA in GC, the most investigated technique to prove the presence of ctDNA is detection of hypermethylation of gene promoters which might result in an inappropriate silencing of tumor suppressor genes [62,64]. The promoter methylation of *APC* and *RASSF1A* in cfDNA was described as frequent epigenetic events in patients with early operable GC [65]. Aberrant methylation of other genes such as *PCDH10, SOX17, TIMP3, MINT2* and *WAF1* also showed promising results in GC [64].

In esophageal squamous cell carcinoma (ESCC), preliminary studies suggested the feasibility of ctDNA detection [66]. Luo et al. used exome or targeted sequencing to detect somatic mutations in 11 patients with ESCC and compared ctDNA from pre- and post-surgery plasma [66]. They compared plasma somatic mutations that were also identified in matched tumors and founded that mutant allelic franction (MAF) decreased after surgery [66].

### 3.4. Hepatocellular Carcinoma

The analysis of the mutational landscape of hepatocellular carcinoma (HCC) over 3000 samples in the Catalog of Somatic Mutation in Cancer showed that the most frequent tumor mutations were *TP53* (27%), *TERT* (25%) and *CTNNB1* (18%) [67,68,69]. Using targeted methods to detect these three genes mutations in plasma, ctDNA presence was proven from 20% to 55% of patients with HCC across different studies [68,69,70,71,72]. In one prospective study including 27 patients with proven ctDNA, only 22% of them (6/27) also had matched mutants in tumor tissues, underlying the heterogeneity of HCC [68]. Therefore, single specific molecular alterations do not seem to be sensitive or specific enough to be used as a diagnostic tool in HCC. Moreover, some molecular alterations could be unspecific for HCC, such as *TERT* mutations that were present in plasma for 9% of patients with cirrhosis and without evidence of HCC on imaging [70].

When using NGS techniques with panel of frequently altered genes in HCC, ctDNA detection rate reached 63% in a prospective cohort of 30 patients, with two thirds of patients with stage A according to the Barcelona Clinic Liver Cancer score (BCLC A). In this study, the concordance rate between plasma and tissue biopsy was 81% [73].

Despite the utility of gene point mutations, DNA methylation seems to be more broadly informative in HCC. In a recent study, a combination of five aberrant methylation biomarkers was able to distinguish HCC samples from control cirrhotic and not cirrhotic tissue samples, with a specificity of 95% [74].

Some single aberrant methylation genes have shown high concordance rates between plasma and tissue in HCC [75,76,77]. Among patients with hypermethylation of *CDKN2A*, which is described in up to 73% of HCC patients, Wong et al., reported a concordance rate of 81% between plasma and tissue biopsy with a specificity of 100% among control patients [75]. Hypermethylation of *RASSF1A* promoter could also to be a candidate and was found in up to 90% of HCC tissues [78,79,80,81,82]. However, it seems to be also detected in patients with non-malignant liver tumor, such as liver cirrhosis, chronic hepatitis B or in healthy controls, with a lower rate (13%, 4%, and 4%, respectively) [82]. Other single hypermethylated candidates, such as *SEPT9, VIM, FBLN1, TFPI2, TGR5, MT1M, MT1G, APC, SPINT2, SFRP1, GSTP1*, or hypomethylated candidates such as *LINE-1* showed promising results for HCC screening [79,83,84,85,86,87,88].

More recently, whole methylome analysis allowed discovering novel methylated DNA markers in HCC. Creation of a new panel with 6 methylated biomarkers (*HOXA1, EMX1, AK055957, ECE1, PFKP* and *CLEC11A*) was able to detect 75% of BLCL 0 and 93% of BCLC B HCC patients meeting Milan criteria and was superior to AFP [74].

### 3.5. Other GI Cancers

Molecular landscape of cholangiocarcinoma (CC) has been widely studied in the past few years trying to detect therapeutic targets [89,90,91]. The cholangiocarcinoma (CC) is usually separated between intrahepatic CC (IHCC) and extra hepatic CC (EHCC). Some mutations such as *KRAS, BRAF* or *TP53* are more frequent in EHCC but remain rare, whereas others, such as *FGFR1-3* fusions and *IDH1/2* mutations are preferentially detected in IHCC and occur in around 15–20% of tumors [89,90,91]. A recent study including 24 CC patients has reported a concordance rate of 74% between mutations in tumor tissue and ctDNA. When stratifying on tumor localization, concordance rate was 92% for IHCC, but only 55% for EHCC [92].

In squamous cell carcinomas of the anal canal (SCCA), Human papillomavirus (HPV) is found in 90% [93]. Therefore HPV DNA appears as the best candidate to assess the presence of ctDNA in SCCA and can be detected in plasma by ddPCR with sensitivity up to 93% in HPV positive-cancers [94]. In a recent study enrolling 8 SCCA patients, ddPCR demonstrated 100% of specificity for the detection of HVP ctDNA [95].

In gastrointestinal stromal tumors (GIST), mutations of exons 9, 11, 13 and 17 of *KIT*, and of exons 12, 14 and 18 of *PDGFRA* are key drivers of oncogenesis and are present in around 85–90% of tumors, whereas the remaining 10–15% of these cancers is referred as *KIT/PDGFRA* wild-type GISTs. However, other genes, such as *BRAF, NF1*, and *SDH*, may be aberrant in this context [96,97,98]. Therefore, the majority of the studies evaluating the utility of ctDNA in GIST were focused on *KIT* alterations. The ctDNA detection rate varies from 45% to 55% across studies [99,100]. Among 102 archival tumor tissue samples and 163 plasma samples at baseline from patients included in the phase III GRID study in patients with GIST treated by regorafenib versus placebo, following failure of at least imatinib and sunitinib, Demetri et al. first reported a concordance rate of 100% and 71% between plasma and tissue biopsies for primary *KIT* exon 9 and 11 mutations, respectively [101]. Concordance rate also depended on the primary or secondary character of the mutation. For primary *KIT* mutations concordance rate between tissue and plasma was 84% whereas secondary *KIT* mutations were more often detected in plasma (47%) than in tissue (12%) [101]. More recently, Arshad et al. revealed a concordance rate for detection of mutations in GISTs with a positive predictive value of 100% among 243 patients [99].

## 4. Minimal Residual Disease and Detection of Early Recurrence

Detection of minimal residual disease (MRD) is an important challenge as MRD might serve as a surrogate marker for disease free survival (DFS) and could therefore guide further therapeutic interventions for patients after curative treatment. In GI cancers, assessment of MRD through the analysis of ctDNA is not standardized yet but has already been evaluated in different studies. The most relevant of them are summarized in Table 1 and Table 2.

### 4.1. Colorectal Cancer

Around 50% of patients with localized (stage II–III) and resected CRC will further develop metastasis [113]. The addition of adjuvant chemotherapy significantly lowers the risk of relapse [114]. The clinical utility of tracking ctDNA to detect MRD and stratifying patients based on their risk of developing relapse has now been well established in CRC [102,103,104,105,106,115,116].

For stage II colon cancer, Tie et al. described a correlation between disease recurrence and the levels of ctDNA in post-surgery setting. In a prospective cohort of 230 patients, ctDNA was detected in 7.9% (14/178) after surgery. Among them, after a median follow –up of 27 months, 79% (11/14) presented a tumor relapse. However, relapse occurred with a very lower frequency of 9.8% (16/164) in patients with negative ctDNA. For patients who completed adjuvant chemotherapy, the recurrence-free survival (RFS) was less frequent when ctDNA was undetectable treatment. This data present ctDNA as a relevant method to assess presence of residual disease after stage II colon treatment. It could therefore be used to identify patients at higher risk of recurrence [102].

In stage III colon cancer, the same team reported that ctDNA detection in 21% (20/96) of postsurgical samples was also associated with inferior RFS. The ctDNA was detected in 17% (15/88) of post-chemotherapy samples. The estimated 3-year recurrence free interval (RFI) was significantly lower when ctDNA was detectable. Postsurgical ctDNA status was an independent prognostic factor of RFI [103]. Taieb et al. also worked on the predictive value of ctDNA in adjuvant setting in stage III colon cancer in a large series from patients of the IDEA-FRANCE trial (NCT-00958737) [105]. The aim of this analysis was to determine the prognostic and predictive value of ctDNA for adjuvant treatment duration lasting 3 versus 6 months of oxaliplatin-based adjuvant chemotherapy. The samples of 1017 patients were fully analyzed for ctDNA detection. Among them, 140 patients (13.8%) had ctDNA-positive samples after surgery. The 3-years DFS rates were 66.4% for positive versus 76.7% for negative-ctDNA samples, respectively. Multivariate analysis confirmed that the presence of ctDNA was as an independent prognostic marker. In this series, 6 months of chemotherapy showed better results than 3 months in both ctDNA-positive and –negative groups. Interestingly, ctDNA-positive patients treated for 6 months and ctDNA-negative patients treated for 3 months had a similar prognosis. [105].

In locally advanced rectal cancer (T3/T4 and/or N+), Tie et al. reported in another prospective study that ctDNA could detect MRD after chemoradiotherapy or surgery. The ctDNA detection after these treatments were correlated with an increased risk of recurrence, and a shorter 3-years RFS (33% versus 87% for patients with positive and negative post-operative ctDNA, respectively) [106].

In a larger series, Tarazona et al. recently evaluated the detection and longitudinal monitoring of ctDNA in CRC patients pre- and post-operatively, during and after adjuvant chemotherapy in a prospective multi-centric study on 193 patients with resected stage I-III tumors. Among the 14 out of 152 (9.2%) patients with post-operative ctDNA before adjuvant chemotherapy (identified to be MRD-positive), 78.5% (11/14) relapsed. In contrast, 10.1% (14/138) of MRD-negative patients relapsed. In the multivariable analysis, longitudinal ctDNA status was the only significant prognostic factor associated with RFS. Serial ctDNA analysis also allowed detecting MRD up to a median of 9.08 months before radiologic relapse, with a sensitivity of 79.1% and specificity of 99% [104].

Conventional surveillance strategy might therefore be completed by ctDNA analysis to stratify the risk of recurrence and guide therapeutic interventions in CRC. Interventional trials to assess the clinical benefit of the monitoring of ctDNA in adjuvant setting are currently ongoing in several countries, such as PRODIGE 70-CIRCULATE trial (NCT-04120701). The aim of this French multicentric study is to identify, through the detection of ctDNA, a group of patients with higher risk of relapse among patients with stage II operated colon cancer and to test the benefits of adjuvant chemotherapy (FOLFOX6m-5fluorouracile, leuvocorin and oxaliplatin) in this population.

Recently, ctDNA was also used to detect MRD in post-operative setting in patients with oligometastatic CRC. Among 100 patients in this study, MRD-positive status was associated with a higher level of relapse and also with an inferior overall survival (OS) [107].

### 4.2. Pancreatic Cancer

In early-stage PC, there is currently no biomarker to guide adjuvant treatment. Several studies and meta-analysis already described the negative prognostic value of ctDNA in localized PC at baseline or in post-operative setting [41,44,108,117,118,119,120,121]. In pre-operative setting, a recent meta-analysis of 5 retrospective studies including 375 patients reported that pre-operative ctDNA detection was significantly associated with poor OS and with a trend to higher risk for disease recurrence [120]. After surgery, immediate post-operative ctDNA detection was associated with a trend for poorer RFS and with a significant poorer OS [120]. These data are in line with the results of some prospective studies including patients with early stage PC [44,119,121].

However, ctDNA is currently unlikely to become a routine tool to avoid some adjuvant treatment in PC due to the high recurrence rate after surgical resection. However, it may help to detect early relapse and therefore shorten the time to treatment. In some resected PC, longitudinal ctDNA monitoring allowed detecting MRD up to a median from around 2.7 to 6.5 months before radiologic relapse [109,122]. However, the potential clinical benefit of early-relapse treatment based still need to be evaluated.

### 4.3. Esophageal and Gastric Cancer

There are few data on the prognostic and predictive value of MRD in context of esophageal and gastric cancer.

Focusing on 29 patients with resectable GC in their cohort, Maron et al. recently found that patients with detectable ctDNA prior to surgery/therapy had a trend to shorter DFS than those with undetectable ctDNA [63]. Interestingly, after surgery, the residual detection of ctDNA was significantly correlated with worse outcome [63]. However, three apparently positive ctDNA after surgery did not relapse and the mutations detected were not present in their tissue analysis and should therefore trend to a cautious interpretation of these results [63].

In a recent prospective study enrolling 35 patients with localized esophageal adenocarcinoma and 10 patients with ESCC, Azad et al. showed that the detection of ctDNA by deep sequencing method after exclusive or preoperative chemo-radiotherapy was associated with disease progression, formation of distant metastases, and shorter DFS. Moreover, detection of ctDNA after exclusive or preoperative chemoradiotherapy anticipated by 2.8 months radiographic tumor progression [110]. These results are in line with those from a former study showing that ctDNA increased approximately 6 months earlier than the detection of tumor recurrence by imaging tests in two patients with ESCC [123]. In another retrospective study among 17 ESCC patients with stage IIA to IIIB tumor, cfDNA was screened pre and post-surgery. Among the 8 patients with somatic mutations detected in plasma, corresponding to ctDNA in pre-surgery, only 2 patients still had these mutations detected in post-surgery setting and with a lower MAF, suggesting that ctDNA could potentially be used to monitor disease load and detect MRD [124].

### 4.4. Hepatocellular Carcinoma

After surgery in HCC, the detection of ctDNA, proven by the detection of tumor specific alterations and by the use of methylation panel, seems to be correlated with worse prognosis in several small studies [69,70,75,80,110,111,125,126,127,128].

Recently, a larger retrospective series on 81 resectable HCC showed shorter DFS and OS in patients with presence of *TERT, CTNNB1* or *TP53* mutations in plasma after curative hepatectomy [111]. In a multivariate analysis of this study, detectable ctDNA was the only independent risk factor for postoperative recurrence [111].

Regarding the use of methylation markers, a first study has reported that higher methylation of *RASSF1A* in plasma at diagnosis or one year after surgery was correlated with poorer DFS in 63 patients with resectable HCC. However, higher methylation rate one month after resection was not correlated with significantly shorter DFS [80]. Similar conclusions have been reported with the detection in plasma of hypermethylation of insulin growth factor binding protein-7 among 155 HCC patients after surgical resection. This detection was shown as an independent prognostic factor for poorer OS and higher tumor recurrence [127]. In advanced HCC, combination of *LINE-1* hypomethylation and measurement of *RASSF1A* hypermethylation were also described as correlated with poorer OS, earlier recurrence and with poorer prognosis upon curative resection [129].

### 4.5. Other GI Cancers

In locally advanced SCCA, Cabel et al. recently used ddPCR to detect HPV ctDNA in a study of 18 patients. In this series, presence of HPV ctDNA after chemoradiotherapy was associated with a poor prognostic. In most patients, HPV ctDNA was detectable before chemoradiotherapy and became undetectable along treatment. Patients with residual ctDNA after completed treatment had a shorter DFS [112]. A trial evaluating the detection of ctDNA after curative treatment in patients with pelvic stage II–III HPV induced cancer is currently ongoing (Circa HPV study—IC 2017-01).

In localized GISTs, ctDNA might be used as a tumor specific biomarker for early prediction of recurrence, as suggested by Maier et al. who first described that the amount of ctDNA correlates with prognosis [130]. However, few data are currently available in this context. In pre-operative setting, Kang et al. recently demonstrated that ctDNA could be used as a surrogate biomarker for tissue biopsy to determine *KIT* and *PDGFRA* mutations among 25 patients with GISTs [131]. However, due to the small effective, these results should be further validated in larger series.

## 5. Circulating Tumor DNA as Tumor Burden and/or Prognostic Marker in GI Cancers

The probability to detect ctDNA in GI cancers is associated to tumor stage. Bettegowda et al. analyzed the presence of ctDNA in various cancer types among 640 patients. Concerning GI cancers, they reported the presence of ctDNA in 73%, 57% and 48% of patients with localized CCR, esogastric cancers and PC, respectively, whereas ctDNA was detectable in more than 75% of patients with advanced cancers [132].

### 5.1. Colorectal Cancer

In CRC, ctDNA appears strongly correlated with tumor burden according to several studies [133,134]. Tie et al. described that pretreatment ctDNA levels correlated more strongly with initial tumor burden as estimated from standard RECIST 1.1 criteria than pretreatment CEA [133]. Moreover, ctDNA levels were described to be significantly associated with presence of liver metastasis and sum of the tumor diameter in metastatic sites [32,134]. However, the association between ctDNA levels and lung, lymph node and peritoneal metastasis, tumor markers, primary tumor location, and number of metastatic organs in CRC remains more controversial [133,134,135].

Furthermore, numerous studies have suggested that detectable ctDNA at diagnosis was strongly negatively correlated with progression-free survival (PFS) and OS, regardless of other prognostic factors or detection method and sample type [136,137,138,139,140,141].

### 5.2. Pancreatic Cancer

In PC, several studies are in line with the results of Bettegowda et al. showing that ctDNA detection increases with tumor stage and reflects tumor burden [44,142]. In locally advanced or metastatic PC, the prognostic role of ctDNA has been largely evaluated. The presence of ctDNA at baseline before first-line treatment appears to be correlated with worse survival in many studies and meta-analysis [42,44,142,143,144]. Relation between quantitative detection of *KRAS* mutations and prognosis is however still unclear in PC.

### 5.3. Esophageal and Gastric Cancer

In GC, a recent meta-analysis Gao et al. reported a significant association between ctDNA and tumor stage, presence of lymph node and distant metastasis [62]. The authors also reported that high level of ctDNA in GC was associated with worse OS [62]. The largest ctDNA dataset yet published, analyzed the serum of 1630 patients with GC to detect ctDNA through NGS method. These authors then recently confirmed that the maximal tumor somatic variant allelic frequency, defined as the largest mutated ctDNA clone detected among all cfDNA present in the plasma that is used to estimate overall ctDNA quantity appeared as a surrogate biomarker for disease volume or burden in metastatic disease [63].

In metastatic GC, it seems that a higher amount of ctDNA at baseline is associated with worse outcomes in the cohort of Maron et al. [63]. Because of the lack of data on the prognostic and predictive impact of ctDNA in GC, an ongoing observational prospective study (PLAGAST) is currently evaluating the correlation between the level of ctDNA and prognosis or response to treatment of localized and advanced gastric cancer (NCT-02674373).

In esophageal squamous cell carcinoma (ESCC), one small study described that the MAF in ctDNA changed concomitantly with tumor burden in two patients [123].

### 5.4. Hepatocellular Carcinoma

In HCC, ctDNA detection rate was higher in metastatic tumors and correlated with tumor burden [73,145]. For patients with unresectable advanced HCC, ctDNA levels showed a significant correlation with the presence of metastases and survival in an exploratory cohort of 13 patients recruited in the context of the SORAMIC trial [146]. Moreover, in a prospective study among 41 patients with HCC, Liao et al. reported that the presence of *TERT, CTNNB1* or *TP53* mutations in plasma before surgery were significantly associated with shorter RFS [69]. Lastly, the amount of ctDNA detected immediately after local treatment was significantly correlated with the presence of distant metastases, supporting also a potential prognostic value [146].

### 5.5. Other GI Cancers

In CC, one study recently described a significant correlation between ctDNA MAF in both IHCC and EHCC, with the respective initial tumor load [92]. Furthermore, in the study of 24 CC patients, ctDNA variant allele frequency values at baseline showed a trend for a shorter PFS but this correlation was significant when focusing on the IHCC group [92].

In SCCAs, the median level of ctDNA was higher in metastatic than in localized tumor stage in two recent studies [112,147]. Bernard-Tessier et al. described in an ancillary study of 57 SCCA patients that ctDNA level has a significant impact on clinical outcomes. The PFS was significantly longer for patients with ctDNA level below the cut-off obtained by area under the curve (AUC) [147]. In this study, HPV ctDNA negativity in non-progressing patients after chemotherapy completion was also a strong predictive biomarker of extended response to chemotherapy [147].

In GISTs, the amount of ctDNA was reported as significantly higher in patients with an active disease compared to those in remission [130]. In a recent study among 44 unpretreated GISTs, ctDNA was detected in all patients with metastatic disease whereas it was inconstant in patients with localized tumor. In this study, tumor burden was the most important detection determinant [148].

## 6. Circulating Tumor DNA to Monitor Treatment Response and Detect Acquired Resistance

The non-invasive nature of ctDNA allows for repeated testing and molecular assessment of tumor during treatment. This dynamic assessment is a clear advantage over traditional tissue biopsy. In the advanced tumor stage, baseline ctDNA could be more helpful to capture the molecular spatial and temporal heterogeneity of the disease which is a particularly important biological issue, at diagnosis or later because of clonal evolution and selection [33]. Differences in molecular characteristics have been described between primary tumor and metastases, especially in metachronous lesions [34].

Moreover, the monitoring of ctDNA may also anticipate the evaluation of treatment efficacy by detecting emergent actionable molecular alterations implicated in therapeutic resistance to ongoing treatment.

### 6.1. Colorectal Cancer

In mCRC, longitudinal quantification of ctDNA appears to be correlated with tumor evolution in several studies [133,140]. By sequencing a panel of 15 genes with frequent somatic variant in CRC tissue sample at diagnosis of 53 patients with mCRC, Tie et al. evaluated ctDNA as disease monitoring. They reported that a level of reduction in ctDNA concentration during first cycle of chemotherapy was significantly associated with the objective radiologic response rate at 8–10 weeks and with a trend for a better PFS. [133]. Similarly, Garlan et al. showed that early changes of the ctDNA concentration could predict the efficacy of first- or second-line chemotherapy in a prospective cohort of 82 mCRC. They used ctDNA monitoring between the first and second or/and third cycle of chemotherapy to define a composite marker that allowed to separate patients in two groups of “bad” or “good” ctDNA responder. This marker was based on the “normalization” of the ctDNA concentration (thresholds of 0.1 ng/mL) and the slope of ctDNA concentration decrease. The group of better ctDNA responders demonstrated a significantly better tumor response rate, PFS and OS [140]. The changes of ctDNA concentration during treatment therefore appear as a relevant early tool to assess treatment efficacy and this biomarker should be evaluated in larger prospective series.

Furthermore, ctDNA can also be used to track clonal evolution. It has been suggested that CRC presumably contains resistant mutant clones before treatment that emerge under therapeutic pressure [149]. The acquisition of resistance can be accompanied by the emergence of *RAS* pathway mutations that could allow to anticipate radiologic progression [150,151]. Several studies have already described emergence of mutations detected by ctDNA under anti-EGFR treatment up to 5–10 months before imaging diagnostic [150,151,152]. By monitoring ctDNA, Siravegna et al. also showed in a subset of patients, that the proportion of ctDNA, based on the detection of *KRAS* mutations, dynamically varied depending on the presence or the absence of anti-EGFR treatment. These possible dynamic clonal evolutions induced by therapeutic pressure justified to re-challenge anti-EGFR based treatment after a withdrawal period in mCRC. [152,153]. Some retrospective analyses of the phase 2 CRICKET and E-Rechallenge studies suggested that ctDNA could guide this re-challenge therapy because only patients without RAS or BRAF circulating mutations detected plasma at the time of re-challenge might achieve clinical benefit from the retreatment with anti-EGFRs [116,153,154] (Table 3).

More recently, Sartore-Bianchi et al. presented the results of the CHRONOS study, the first interventional ctDNA guided study in mCRC. 52 patients were screened by liquid biopsy for anti-EGFR re-challenge. Among them, 16 (31%) were mutated in ctDNA for *RAS*, *BRAF* or *EGFR* ectodomain and avoided a useless treatment by anti-EGFR. Of the 36 (39%) triple wild-type patients, 27 were re-challenged by anti-EGFR and obtained an ORR of 30% [155]. Some ongoing studies, such as the prospective RASINTRO study (NCT-03259009) or the randomized FIRE4 trial (NCT02934529) are currently ongoing to confirm the clinical use of liquid biopsy-driven re-challenge and the predictive impact of *RAS* mutations in ctDNA for the efficacy of anti-EGFR reintroduction treatment in patients with mCRC (Table 3).

Other molecular alteration than *KRAS* mutations can emerge under therapeutic pressure and can be detected by analysis of ctDNA, such as amplifications of *MET* and *HER2* or *EGFR* mutations [156,157,158].

### 6.2. Pancreatic Cancer

In advanced PC, some regimens such as FOLFIRINOX (5-fluorouracil, leucovorin, oxaliplatin, and irinotecan) and gemcitabine plus nab-paclitaxel are effective but are not devoided of toxicities [159,160,161,162]. The monitoring of *KRAS* mutation through ctDNA has been performed in several studies and suggested that its detection could predict radiological progression, but some results were however discordant [119,163,164]. The clearance of *KRAS* ctDNA during treatment predicted better PFS than remaining positive ctDNA [164], and increasing levels of *KRAS* ctDNA were also associated with worse PFS and OS [165,166]. Finally, the decline slope of ctDNA concentration based on mutation of *KRAS* was associated with OS in another study [167]. Apart from *KRAS* mutations, evolution of other mutations in plasma, such as *TP53, SMAD4, CDKN2A, KRAS, APC, ATM, FBXW7* and others could also be used to reliably reflect response to therapy [47,48].

Unlike other GI cancers, there is currently no targetable molecular alteration for all patients with advanced PC in clinical routine. However, some new treatment could be promising in PC, such as PARP inhibitors in case of germline *BRCA1/2* mutations [168]. Moreover, like in other tumors, checkpoints inhibitors seem to be efficient in advanced PC with microsatellite instability [169,170,171]. Molecular alterations could be detected in ctDNA in PC [163] and therefore maybe screen patients for targeted therapies in the future. In this context, Bachet et al. recently confirmed from the data of a randomized phase II trial that the ctDNA could be a predictive biomarker of l-asparaginase encapsulated in erythrocytes (eryaspase) efficacy in advanced PC [172].

### 6.3. Esophageal and Gastric Cancer

In patients with advanced gastroesophageal adenocarcinoma, the addition of trastuzumab to chemotherapy was associated with improvement of clinical outcomes for tumors with a high level of HER2 expression (IHC3+ or IHC2+ and FISH+) [173]. Some studies have already described the potential for ctDNA to detect *HER2* amplification by ddPCR with high concordance with classic immunohistochemistry and fluorescent in situ hybridization on tissue samples [174,175]. However, in the recent cohort of Maron et al. seven patients with advanced disease were tested for *HER2* amplification in both primary and metastatic tumor, and in ctDNA. Among them, only 2 patients (28%) were concordant for *HER2* amplification detection in the three samples, underlying possible missed detection of *HER2* amplification by NGS and then the risk of missed opportunities to use anti-HER2 therapies [63]. Despite its lack of sensitivity, ctDNA could however be used in combination with tissue NGS to define a group of extremely sensitive *HER2* amplified patients when treated with trastuzumab [63].

Moreover, some authors already suggested that ctDNA could also be used to monitor response to therapy in GC. In a recent study, tumor responses to lapatinib plus capecitabine were closely related with changes of the level of amplification of *HER2* detected in plasma through serial ctDNA sequencing [176]. In the study of Maron et al. dynamic measurements of ctDNA before and during treatment showed that a decrease superior to 50% in MAF was correlated with better OS [63]. The detection of therapeutic resistance to treatment in advanced GC could also be improved by ctDNA. In the cohort of Maron et al. some anti-HER2 therapy acquired resistance mechanisms were detected using ctDNA [63].

### 6.4. Hepatocellular Carcinoma

In advanced HCC, ctDNA could be used to monitor tumor burden under therapy. A diagnostic prediction model with 10 selected methylation markers through ctDNA was recently developed by Xu et al. and correlated with tumor burden, treatment response, and disease stage [145].

In a study using whole exome sequencing to evaluate ctDNA among HCC patients who underwent surgery, in patients with positive ctDNA after surgery, the levels of serum ctDNA increased with disease progression and responded to the additional treatments [126].

The somatic MAF of ctDNA could also reflect clinical dynamics as demonstrated in one patient with advanced HCC undergoing trans-arterial chemoembolization in whom increasing level of 8 somatic mutations in plasma was detected before imaging diagnosis and the increase of standard biomarker AFP [177].

### 6.5. Other GI Cancers

In CC, until past years, chemotherapy was the only validated treatment for advanced disease [91,178]. Recently, some targeted therapies emerged in the therapeutic arsenal. Ivosidenib, a first-in-class oral IDH1 inhibitor, has demonstrated an improvement of PFS over placebo in advanced CC with *IDH1* mutations in the phase III ClarIDHy study [179]. In another phase II study (NCT-02150967), BGJ398, an orally bioavailable, selective pan-FGFR kinase inhibitor demonstrated clinical activity against chemotherapy-refractory CC with *FGFR2* fusions [180]. Lastly, the phase II study FIGHT-202 also supported the efficiency of pemigatinib, an oral inhibitor of FGFR1, 2, and 3 in previously treated patients with cholangiocarcinoma with *FGFR2* fusions or rearrangements [181]. Therefore, the interest in monitoring ctDNA in CC is increasing. Goyal et al. already monitored 9 patients with *FGFR2* fusions and detected de novo point mutations that conferred resistance to BGJ298 in all patients (*n* = 3) who underwent progression [182]. Ettrich et al. recently demonstrated that 63% of treatment naïve patients with advanced CC had changes in their mutational profile during chemotherapy. They evaluated and identified a set of 76 potential progression driver genes among a large-scale panel sequencing of 710 cancer-related genes [92]. These data suggest that ctDNA could be used to track disease progression.

In SCCA, few data suggest that HPV ctDNA could be used to monitor the efficacy of immunotherapy as suggested in a recent case report [183].

In GIST, one main application of ctDNA seems to be monitoring response to therapy and tracking therapeutic resistance to tyrosine kinase inhibitors (TKI) [99,131,148,184,185]. Indeed, despite the revolution in GIST management through the contribution of first line TKI such as imatinib targeting *KIT* or *PDGFRA* molecular drivers, the majority of GIST will progress with the acquisition of secondary *KIT* or *PDGFRA* mutations. In this context, second and third line TKI have been used in some refractory GIST patients [186,187,188,189]. Maier et al. first described a dynamic change in MAF in plasma of advanced GIST under treatment. A decrease or a disappearance of ctDNA occurred in patients responding to TKIs [130]. In other studies, the usefulness of ctDNA for the identification of TKI resistance mutations and their prognostic utility was demonstrated [184,185]. In a phase II study patients with secondary *KIT* mutations had significantly worse OS than those with no detectable secondary mutations [184]. ctDNA can also be used to detect resistance mutations in other gene than *KIT*, as demonstrated in a prospective study that collected 30 plasma samples from 22 patients with metastatic GIST [190]. Monitoring ctDNA using NGS patients with GIST under TKI treatment detected primary but also secondary mutations emerging in patients who had a progressive disease whereas only primary mutations were detected in patients with stable disease. These resistance mutations in ctDNA could represent early biomarkers for treatment response [148,191].

## 7. Conclusions

CtDNA harbors great potential to improve the management of patients with GI cancers. However, the level of advancement of its development in the different tumor types is still inhomogeneous. It seems that the level of data for colorectal cancer might soon allow the use the ctDNA in clinical routine both in adjuvant and in metastatic setting. In pancreatic cancer, the level of proof could be soon reliable but the applications remain more limited for the moment, partially due to the dark prognosis of these tumors and the lack of efficient therapeutic arsenal. In GIST as for CRC, due to the amount of resistance mutations and the frequent use of tyrosine kinase inhibitors, ctDNA appears as a very promising tool. For most other gastro-intestinal cancers, ctDNA shows promising preliminary data but further studies are still needed, and some of them are currently ongoing, to help specifying the exact role of ctDNA in clinical routine.

Lastly, the ctDNA might also be used as a surrogate marker to predict a response to immunotherapy. The mismatch repair deficient (dMMR) or microsatellite instability (MSI) status cannot be directly assessed by liquid biopsy. However, the tumor mutation burden can be determined on plasma and appears to be strongly correlated with dMMR/MSI high status and tumor response to immune checkpoint inhibitors. It is strongly relevant as ICI are becoming a key factor in management of GI cancer and particularly in metastatic CRC cancer in which pembrolizumab recently became the first line reference treatment in MSI-high metastatic CRC.

## Figures and Tables

**Figure 1 cancers-13-04743-f001:**
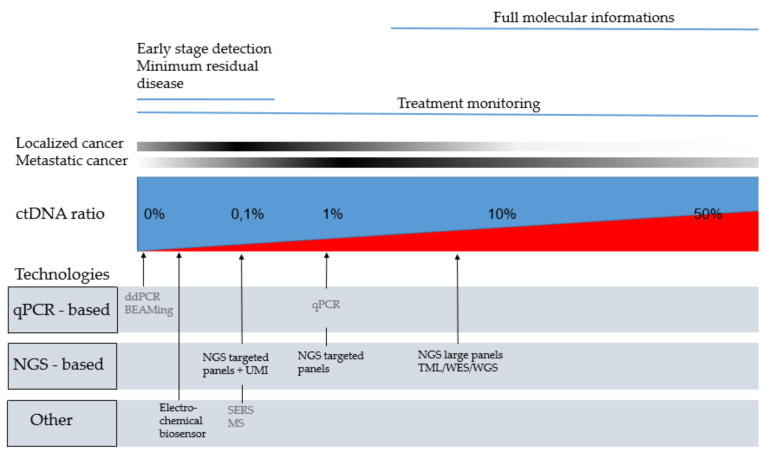
Technologies that can be use regarding ctDNA ratio and the clinical application. In grey: technologies with limited targets suggesting prior knowledge of the mutation; In black: technologies with broad genes coverage; In blue: non tumoral cell free DNA. In red: circulating tumor DNA; BEAMing: beads, emulsion, amplification, magnetics; ctDNA: circulating tumor DNA; ddPCR: digital droplet PCR; MS: Mass spectrometry; NGS: new generation sequencing; qPCR: quantitative PCR; SERS: Surface-Enhanced Raman Scattering; TLM: tumor mutation load; UMI: unique molecular identifiers; WES: whole exome sequencing; WGS: whole exome sequencing.

**Table 1 cancers-13-04743-t001:** Main Minimal Residual Disease assessment studies for circulating tumor DNA in colorectal cancers.

Reference	Type of Study	Tumor Location	Stage (TNM)	MRD Assessment Method	Number of Pts with ctDNA + after Surgery (%)	RFS/DFS after Surgery	HR for Relapse; *p*-value	% of pts Who Underwent ACT	Number of pts with ctDNA + after ACT (%)	RFS/DFS after ACT	HR for Relapse; *p*-Value
Tie et al. [102]	Prospective	Colon	II	PCR (Safe-SeqS)	Patients not treated by ACT: 14/178 (7.9%)	3 years RFS in patients not treated by ACT: in ctDNA –: 90% in ctDNA +: 0%	HR = 18; *p* < 0.001	23%	6/52 (11%)	NA	HR = 11; *p* = 0.001
Tie et al. [103]	Prospective	Colon	III	PCR (Safe-SeqS)	20/96 (21%)	3 years RFS: in ctDNA –: 76% in ctDNA +: 47%	HR = 3.8; *p* < 0.001	100%	15/88 (17%)	3 years RFS: in ctDNA-: 87% in ctDNA +: 33%	HR = 6.8; *p* < 0.001
Tarazona et al. [104]	Prospective	Colorectal	I-III	Personalized ddPCR	14/152 (9.2%)	NA	HR = 16.53; *p* < 0.001	NA	7/25 (28%)	NA	HR = 10.02; *p* < 0.0001
Taieb et al. [105]	Prospective	Colon	II-III	ddPCR	140/1017 (13.8%)	3 years DFS: in ctDNA –: 77% in ctDNA+: 66%	HR = 1.55; *p* = 0.015	100%	NA	NA	NA
Tie et al. [106]	Prospective	Rectum	Locally advanced T3/T4 and or N+	PCR (Safe-SeqS)	19/159 (11.9%)	3 years RFS: in ctDNA –: 87 % in ctDNA +: 33%	HR = 13; *p* < 0.001	64%	NA	NA	NA
Loupakis et al. [107]	Prospective	Colorectal	IV (Oligometastatic)	Personalized and tumor informed multiplex PCR (Signatera)	52/100 (52%)	NA	HR = 4.6; *p* <0.001	38%	NA	NA	NA

ACT: adjuvant chemotherapy; CI: Confidence interval; CT: chemotherapy; ctDNA: circulating tumor DNA; ddPCR: digital droplet PCR; DFS: Disease free survival; HR: hazard ratio; MRD: Minimal residual disease; PCR: polymerase chain reaction; pts: patients; NA: Not available; NGS: New generation sequencing; NR: Not reached; RFS: Recurrence free survival.

**Table 2 cancers-13-04743-t002:** Main Minimal Residual Disease assessment studies for circulating tumor DNA in gastro-intestinal cancers except colorectal.

Reference	Type of Study	Tumor Location	Stage (TNM)	MRD Assessment Method	Treatment	Number of pts with ctDNA + after treatment (%)	RFS/DFS in ctDNA + after Treatment (months)	RFS/DFS in ctDNA-after Treatment (months)	RFS/DFS in ctDNA-vs. + after Treatment HR; *p*-Value
Pietrasz et al. [44]	Prospective	Pancreas	Resectable	ddPCR and targeted NGS	Surgery	6/31 (19.4%)	4.6	17	HR: NA; *p* = 0.03
Nakano et al. [108]	Retrospective	Pancreas	Resectable	Peptide nucleic acid-directed PCR clamping	Surgery +/− neoadjuvant chemotherapy	20/45 (44.4%)	NA	NA	HR = 2.919; *p* = 0.027
Groot et al. [109]	Prospective	Pancreas	Resectable/Borderline	ddPCR	Surgery +/− neoadjuvant chemotherapy	11/41 (26.8%)	5	15	HR: NA; *p* < 0.001
Maron et al. [63]	Retrospective	Gastric	Resectable	NGS	Surgery +/− neoadjuvant chemotherapy	7/22 (31.8%)	12.5	NR	after surgery: HR = 0.1; *p* = 0.03
Azad et al. [110]	Retrospective	Esophageal	Localized	Cancer personalized profile sequencing (CAPP-seq)	Chemoradiotherapy (+/− surgery)	5/31 (16.1%)	NA	NA	HR = 18.7; *p* < 0.0001
Wang et al. [111]	Prospective	Hepatocarcinoma	Resectable BCLC 0-C	ddPCR	Surgery	17/53 (32.1%)*	7	20.8	HR: NA; *p* < 0.001
Cabel et al. [112]	Prospective	SCCA	Locally advanced-stages II-III	ddPCR	Exclusive chemoradiotherapy	3/18 (17%)	NA	NA	HR: NA; *p* < 0.0001

* increased ctDNA mutant allele frequency postoperatively. CI: Confidence interval; ctDNA: circulating tumor DNA; ddPCR: digital droplet PCR; DFS: Disease free survival; HR: hazard ratio; MRD: Minimal residual disease; PCR: polymerase chain reaction; NA: Not available; NGS: New generation sequencing; NR: Not reached; RFS: Recurrence free survival; SCCA: Squamous cell carcinoma of the anal canal; pts: patient.

**Table 3 cancers-13-04743-t003:** Studies evaluating circulating tumor DNA as a screening tool to detect patients who could benefit from anti-EGFR re-challenge in metastatic colorectal cancer.

Reference/ NCT	Type of Study	Status	Detection Technique	Mutations Analyzed	Primary Outcoume	Secondary Outcomes	Number of Patients Evaluated	Mutational Status at Rechallenge - Number (%)		Number of Previous Treatment Line(s)	PFS (Months) According to Mutational ctDNA Status		HR; *p*-Value
								Wt	Mutated		Wt	Mutated	
Cremolini et al. [153]	Multicenter phase II single arm	Achieved	ddPCR	*RAS*	ORR	PFS and OS	25	13 (52%)	12 (48%)	2	4.0	1.9	HR = 0.44; *p* = 0.03
Sartore-Bianchi et al. [155]	Multicenter interventional phase II	Achieved	ddPCR	*RAS, BRAF, EGFR*	ORR	PFS and OS	52	36 (69%)	26 (31%)	2–6	4.0	Not treatead by anti-EGFR	NA
Nakamura et al. [154]	Multicenter phase II single arm	Achieved	dPCR	*KRAS, NRAS, BRAF,* *PIK3CA, EGFR S492R*	RR	PFS, OS, aEFI	33	NA	NA	NA	7.0	2,9	NA
NCT-03259009 (RASINTRO)	Prospective observational cohort	Recruitment achieved	NGS	*RAS*	PFS	Tumor response and OS	73 (estimated)	-	-	-	-	-	-
NCT-04509635	Single center Prospective interventional randomized	Not yet recruiting	NA	*RAS*	DCR	ORR, PFS and OS	50 (estimated)	-	-	-	-	-	-
NCT-04775862	Prospective phase II	Recruiting	NA	*RAS*	ORR, PFS	Proportion of *RAS* wt patients after 2nd progression and prevalence of *RAS* G12C mutation	60 (estimated)	-	-	-	-	-	-

aEFI: anti-EGFR antibody free interval; CI: Confidence interval; ctDNA: circulating tumor DNA; DCR: disease control rate; ddPCR: digital droplet PCR; HR: hazard ratio; NA: Not available; NGS: New Generation Sequencing; ORR: Objective response rate; OS: Overall survival; OS1: Overall survival after first line of treatment; PCR: polymerase chain reaction; PFS: Progression free survival; RR: response rate; wt: wild type.

## Data Availability

Not applicable.

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
