# Peer review of "Role of Circulating Tumor DNA in Gastrointestinal Cancers: Current Knowledge and Perspectives"

_cancers, 2021, doi:10.3390/cancers13194743_

Round 1

Reviewer 1 Report

Moati et al. in this excellent review present the role of CtDNA in GI cancers. In addition, they highlight its very promising potential in diagnostic and prognostic settings. This reviewer has suggestion for few textual changes.

  1. Simple summary. Line 13: Please write ‘Treatment (or management) of gastrointestinal cancers’
  2. Simple summary. Line 14: “ in what extend” should be “to what extent”
  3. Simple summary. Line 16: Not sure what is conveyed by ‘it appears as a very promising implement to classic? diagnostic, prognostic’.
  4. Line 38: ‘increase in inflammatory (diseases), infections and cancers’
  5. Section 2. Line 57: Please write ‘histological assessment’ in place of ‘histological tissues assessment’
  6. Section 2. Lines 73-6: Please include the appropriate reference in support of the statement.
  7. Section 2.2. Line 114: Please write ‘the use of highly sensitive …’ in place of ‘the use highly sensitive …
  8. Section 5.1. Line 478: ‘By monitoring ctDNA, Siravegna et al. also shown’ – please write ‘By monitoring ctDNA, Siravegna et al. also showed’
  9. Section 5.1. Line 481: ‘allowed to the re-challenge with EGFR’. Please write ‘allowed to re-challenge’

Author Response

Dear reviewer,

We thank you for your time and comments about our manuscpript.

You will find a revised version of our work in which we corrected the textual changes you underlined.

Moreover, we added a new section and a new figure about molecular aspects of ctDNA uses and current development and with a supplementary.

We hope that this new version will suit you.

Best regards,

Reviewer 2 Report

There is not any novelty in the manuscript. There might only have some updates in comparison with other scientific documents like https://pubmed.ncbi.nlm.nih.gov/30087854/ and https://pubmed.ncbi.nlm.nih.gov/31139561/.

I would suggest to authors for improving and remarkable of their findings by updating and adding laboratory aspect issues (advantages and disadvantages methods, kind of new molecular methods,…) into a new section within the manuscript. So, maybe their findings could be attractive to other scientists.

Author Response

Dear reviewer,

Thank you very much for your comments about our manuscript.

According to your request, we enriched our manuscript with a division devoted to molecular aspects of ctDNA uses and current development and with a supplementary illustrative figure.

We hope that you will find it relevant and that it will improve the attractiveness of the manuscript.

We appreciate your serious consideration of our manuscript and await future correspondence.

Sincerely,

Reviewer 3 Report

In this review manuscript authors picked very catchy and interesting area that is role of circulating tumor DNA in gastrointestinal cancer. Through this review wanted to explain about cr DNA can be a potential use as a therapeutic purpose including as a biomarker. They categorized this review in different categories like all kind of cancer like colorectal, pancreatic, hepatic etc. They have touched upon the molecular signature of mutation causing the disease and simultaneously talked about the current prospective of ct DNA as a possible therapeutic tool. They collected almost all the potential literature and discussed well in this review. Broadly this manuscript is very interesting as well as informative for scientific communities who work on ctDNA.  My only concern in this review that authors need to put more future perspective as a potential therapeutic use, what I found is lacking. The manuscript is perfect for journal when authors will include in that.

Author Response

Dear reviewer,

Thank you very much for your return and comments.

We appreciate your serious consideration of our manuscript and await future correspondence.

Sincerely,

Reviewer 4 Report

Dear Authors,

Thank you very much for the comprehensive review!

I appreciate your efforts and your work!

I kindly ask you to address some minor issues:

1)

In the sentence "However, the use highly sensitive detection methods of ctDNA might lead to false diagnosis of PC." Lines 114-115 

the word "of" is missing between "use" and "highly".

2)

In the sentence "

The monitoring of KRAS through ctDNA has been performed in several studies and suggested that its detection could predict radiological progression, but some results were however discordant." Lines 511-513

Please clearly state that you meant KRAS mutation and not the mere detection of the gene KRAS.

3)

In section 5.5 the molecularly targeted therapy agent pemigatinib is missing.

Please refer to the phase II trial FIGHT-202 that lead to its approval!

Good luck!

Best wishes!

Author Response

Dear reviewer,

We deeply appreciate your time and comments about our manuscript.

We corrected the minor issues and added the missing reference about FIGHT 202 and pemigatinib.

We also incorporated a section about molecular aspects of ctDNA and an illustrated figure.

We appreciate your serious consideration of our revised version of the manuscript.

Sincerely,